# On the Zero-Shot Generalization of Machine-Generated Text Detectors

**Xiao Pu**
Peking University
puxiao@stu.pku.edu.cn

**Jingyu Zhang**
Johns Hopkins University
jzhan237@jhu.edu

**Xiaochuang Han**
University of Washington
xhan77@cs.washington.edu

**Yulia Tsvetkov**
University of Washington
yuliats@cs.washington.edu

**Tianxing He**
University of Washington
goosehe@cs.washington.edu

## Abstract

The rampant proliferation of large language models, fluent enough to generate text indistinguishable from human-written language, gives unprecedented importance to the detection of machine-generated text. This work is motivated by an important research question: How will the detectors of machine-generated text perform on outputs of a new generator, that the detectors were not trained on? We begin by collecting generation data from a wide range of LLMs, and train neural detectors on data from each generator and test its performance on held-out generators. While none of the detectors can generalize to all generators, we observe a consistent and interesting pattern that the detectors trained on data from a medium-size LLM can zero-shot generalize to the larger version. As a concrete application, we demonstrate that robust detectors can be built on an ensemble of training data from medium-sized models.

## 1 Introduction

Thanks to large-scale pretraining and tuning with human feedback (Ouyang et al., 2022), large language models (LLMs) (Chung et al., 2022; Zhang et al., 2022; Touvron et al., 2023) are now able to follow instructions and generate realistic and consistent texts. A prominent example is the recently developed ChatGPT or GPT4 model (OpenAI, 2023), which when instructed, can write documents, create executable code, or answer questions that require world knowledge. In a lot of scenarios, the machine-generated texts have high quality and cannot easily be distinguished from genuine human texts (Dugan et al., 2022; Gehrmann et al., 2019).

These trends give an unprecedented importance to the detection of machine-generated text (Su et al., 2023; Jawahar et al., 2020; Pagnoni et al., 2022a). A lot of work has been devoted to proposing efficient detection models or algorithms (Mitchell

Code and datasets will be available at https://github.com/SophiaPx/detectors-generalization.

et al., 2023; Kirchenbauer et al., 2023; Zellers et al., 2019). However, in most studies, the detector is tested on the same generator model that it is trained/tuned on.

This study is motivated by an underexplored research question: How will the detector perform on a different generator that it is not trained on? This question is important due to multiple reasons: (1) LLMs are becoming increasingly large and expensive. Some of the most recent models are either too large to fit into a common GPU (e.g., LLaMA-65B) or require payment from the user (OpenAI, 2023), making the collection of training samples difficult. (2) The number of released LLMs is growing rapidly. In a real application scenario, the detector needs to cover a wide range of LLMs (including the ones the detector is not trained on), instead of only one generator.

In this work, we collect generation data from a wide range of LLMs. We then train neural detectors on data from each generator and test its performance on other generators. Our primary findings include: (1) In many cases, detectors can zero-shot generalize to a held-out generator (Figure 1). In particular, we observe an interesting pattern that the detector for the medium version of an LLM can generalize to the larger version. (2) None of the detectors generalizes to all generators, implying that an ensemble of detectors/data is necessary for a wider coverage. (3) As a concrete application, we demonstrate that robust detectors can be built on an ensemble of training data from medium-sized models; Excluding large-versions only leads to a minor drop in performance.

## 2 Methodology

We begin by giving an overview of our experiment structure and establish some notations. This study includes detection of a range of popular LLMs (detailed in §3), and we construct train/dev/test sets for each generator. In §4.1, we train neural detectors on

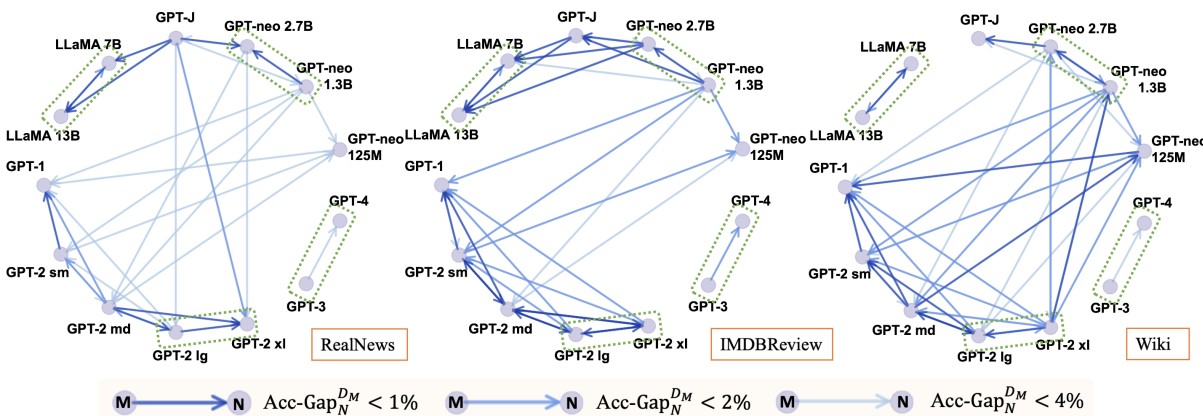

Figure 1: The generalization ability of detectors when applied to different generators, measured by Acc-Gap (defined in §2). Detectors for a medium-size generator can zero-shot generalize to the larger-version model (highlighted by dotted green).

data from each generator and test its performance on other generators. In §4.2, we further consider an ensemble setting, where the detector is trained on data composed of multiple generators, and test its generalization ability on held-out generators.

We denote the detector model trained on data from generator model $M$ as $D_M$. Since we will test the accuracy of $D_M$ on data from different generators, we use $\text{Acc}_N(D_M)$ to denote the accuracy of $D_M$ on the test set of generator $N$. Finally, we define Acc-Gap$_N^{D_M}$ to measure the drop of performance when the detector is trained on generator $M$ instead of $N$ itself:

$$\text{Acc-Gap}_N^{D_M} = \text{Acc}_N(D_N) - \text{Acc}_N(D_M). \quad (1)$$

We expect Acc-Gap to be larger than zero in general, and a large Acc-Gap means $D_M$ has poor generalization on generator $N$.

## 3   Experiment Setup

**Generators**   We include the detection of a total of 13 popular LMs in our study, including GPT-1 (Radford et al., 2018), GPT-2 models (small, medium, large, and xl) (Radford et al., 2019), GPT-3 (text-davinci-003) (Ouyang et al., 2022), GPT-4 (OpenAI, 2023), three GPT-Neo models (125M, 1.3B and 2.7B) (Black et al., 2021; Gao et al., 2020), GPT-J (Wang, 2021), and LLaMA (7B and 13B) (Touvron et al., 2023).[1]

**Datasets**   We consider data from three domains: news, reviews and knowledge. For the news domain, we utilize the RealNews dataset, which

| M | N | RealNews | | IMDBReview | |
|---|---|---|---|---|---|
| | | Gap$_N^{D_M}$ | Gap$_M^{D_N}$ | Gap$_N^{D_M}$ | Gap$_M^{D_N}$ |
| GPT3 | GPT4 | 3.64% | 5.46% | 1.47% | 5.48% |
| LLa7B | LLa13B | -1.11% | 1.18% | -1.50% | 1.47% |
| Neo1.3B | Neo2.7B | 0.04% | 2.16% | -2.31% | 3.41% |
| GPT2lg | GPT2xl | -4.40% | 4.94% | -0.02% | 0.32% |

Table 1: Acc-Gap from medium-version to large-version models based on the ELECTRA detector, as well as in the opposite direction. The generalization from the medium-version model to the large-version is better than the opposite direction.

is a subset of the c4 dataset (Raffel et al., 2019) named "realnewslike". For the reviews domain, we utilize the IMDBreview dataset (Maas et al., 2011). As for the knowledge domain, we utilize the Wikipedia dataset (Foundation)[2].

For each dataset, we first randomly sample 5000 real-world human-written samples, with a train/dev/test split ratio of 8:1:1. For all samples, we truncate the first 20 tokens to serve as prompts and feed them into different generators for text continuation, yielding 5000 machine-generated samples. For generation we apply nucleus sampling (Holtzman et al., 2020) with $p = 0.96$, following the setting in Pagnoni et al. (2022b). We truncate each sample so that its length is around 120 tokens. For all training or test sets in this work, we keep the ratio of human and machine text to be 1:1.

**Detectors**   For data from each generator, we train a ELECTRA-large model (Clark et al., 2020) as a binary classifier. The detectors were trained for 1 epoch with a learning rate of 5e-6 (training

---

[1]There are larger versions of LLaMA, but we find it difficult to fit it into our GPU.

[2]The Wikipedia dataset we used is directly obtained from Hugging Face, data subset "20220301.en" (Page link: https://huggingface.co/datasets/wikipedia).

| | Baseline | | | Pruned (Proposed) | | Pruned (Comparison) | |
|---|---|---|---|---|---|---|---|
| **RealNews Acc(%)** | **Ensemble** | | **Data-mix** | **-GPT4 -L13B** | **-GPT4-GNeo2.7B -L13B-GPT2xl** | **-GPT3 -GPT4** | **-L13B -L7B** |
| | **Vote** | **Prob-avg** | | | | | |
| **Average** | 81.1 | 81.1 | **88.0** | 88.6 (+0.6) | 88.2 (+0.2) | 79.5 (-8.5) | 91.6 (+3.6) |
| **Worst-case** | 50.6 | 50.5 | **84.5** | 84.6 (+0.1) | 83.6 (-0.9) | 42.7 (-41.8) | 80.8 (-3.7) |
| **GPT4** | 50.6 | 50.5 | **88.2** | 84.6 (-3.6) | 85.9 (-2.3) | 42.7 (-45.5) | 93.2 (+5.0) |
| **GPT3** | 52.9 | 52.8 | **87.0** | 87.2 (+0.2) | 87.5 (+0.5) | 52.3 (-34.7) | 91.4 (+4.4) |
| **L13B** | 58.8 | 58.4 | **84.5** | 85.0 (+0.5) | 83.6 (-0.9) | 83.5 (-1.0) | 80.8 (-3.7) |
| **L7B** | 61.6 | 61.3 | **86.0** | 86.1 (+0.1) | 86.1 (+0.1) | 83.7 (-2.3) | 82.6 (-3.4) |
| **IMDBReview Acc(%)** | **Ensemble** | | **Data-mix** | **-GPT4 -L13B** | **-GPT4-GNeo2.7B -L13B-GPT2xl** | **-GPT3 -GPT4** | **-L13B -L7B** |
| | **Vote** | **Prob-avg** | | | | | |
| **Average** | 85.2 | 85.1 | **94.3** | 93.2 (-1.1) | 94.1 (-0.2) | 89.0 (-5.3) | 93.7 (-0.6) |
| **Worst-case** | 52.0 | 51.8 | **93.7** | 92.2 (-1.5) | 92.7 (-1.0) | 62.6 (-31.1) | 89.8 (-3.9) |
| **GPT4** | 52.0 | 51.8 | **94.4** | 93.4 (-1.0) | 94.4 (0) | 62.8 (-31.6) | 94.4 (0) |
| **GPT3** | 54.1 | 54.2 | **93.7** | 93.1 (-0.6) | 93.5 (-0.2) | 62.6 (-31.1) | 93.7 (0) |
| **L13B** | 70.2 | 70.1 | **94.0** | 92.2 (-1.8) | 92.7 (-1.3) | 93.5 (-0.5) | 89.8 (-4.2) |
| **L7B** | 72.1 | 71.9 | **94.1** | 93.0 (-1.1) | 93.9 (-0.2) | 93.7 (-0.4) | 91.7 (-2.4) |

Table 2: Accuracy of the baseline detectors and detectors trained on pruned data. "L13B/7B" refers to the LLaMA 13B/7B generator. We highlight the results for the data-mix model becuase it serves as the base for the pruned models. It is shown that pruning out the large-version LLMs only induce minimal accuracy loss.

for more epochs only gives minimal improvement on the dev set). For the data-mix baseline and pruned models in §4.2, 3 epochs of training is used. We use Adam optimizer (Kingma and Ba, 2014) with $\beta_1 = 0.9, \beta_2 = 0.999$. The average accuracy (when tested on the same generator it is trained on) of all detectors in news, review and knowledge domains are 94.1%, 96.2% and 94.9%, separately.

# 4 Experiment Results

## 4.1 On Generalization Ability of Detectors

As explained in §2, we compute Acc-Gap to reflect the generalization ability of detectors trained on each generator. Figure 1 depicts the Acc-Gap of each detector/generator pair. We link from node $M$ to node $N$ if Acc-Gap$_N^{D_M} < T$ (good generalization), where the threshold $T$ is set to a small number from $\{1\%, 2\%, 4\%\}$. On the other hand, in Figure 3 (Appendix B), node $M$ is linked to node $N$ when Acc-Gap$_N^{D_M} > 20\%$ (poor generalization). For statistical significance, we utilize bootstrapping (Koehn, 2004) and generate 100 virtual test sets by sampling with replacement from the original test set. We then conduct one-sided t-test and use a $p$-value of 0.05.

We observe two interesting patterns shared across the three datasets. First, **the detectors for the medium-version LLMs can generalize to the large-version models.** For example, $D_{\text{LLaMA7B}}$ generalizes to LLaMA13B, and $D_{\text{GPT3}}$ general-

izes to GPT4.[3] This is somewhat surprising because generations from the large-version generator is commonly considered to have higher quality.

Interestingly, the generalization of the reverse direction is weaker on RealNews and IMDBReview. As shown in Table 1, when attempting to generalize from the large-version models to medium ones using ELECTRA detectors, the generalization performance is slightly worse, reflected by a larger Acc-Gap. For the reason behind, we conjecture that comparing to the larger model, the medium generator is making a similar but wider range of artifacts in its generations, leading to a smooth generalization to the detection of the larger model. We also experiment with additional base detectors, e.g. ALBERT Large v2 (Lan et al., 2019) and find that the key observation—that the detectors trained for the medium-size models can generalize to larger-size models—still holds. These results are omitted for brevity.

Second, Figure 3 (Appendix B) shows that **none of the detectors, on its own, can generalize to all generators**. In particular, GPT3 and GPT4 seem "isolated" from other families of generators. This result indicates that if we want an "universal" detector which can cover all generators, an ensemble of detectors/data is necessary. We explore this direction in the next section.

---

[3]Strictly speaking, GPT3 is not a "small version" of GPT4. But they are from the same company, and our experiments consistently show they are strongly related.

## 4.2 Pruning Out Large-Version LLMs in a Mixed Training Dataset

We now demonstrate a concrete application of our findings, and the following realistic threat scenario is considered: The task is still binary classification but the machine text is composed of generations from a range of models (listed in §3). For simplicity, we use a uniform data ratio for the generators.

Following results of the last section, an ensemble of detectors/data is necessary. We begin by comparing two baselines: (1) Model ensemble, where we aggregate predictions from all detectors by majority voting or confidence (probability) average; (2) Data mixing, where we train a new ELECTRA-large detector by mixing up the training data from all generators.[4]

For each baseline detector $D$, we report the average accuracy on all generators, and the worst-case accuracy which is $\min_N \text{Acc}_N(D)$. Accuracy on the four largest generators is also reported. We conduct experiments on RealNews and IMDBReview datasets, and the results for baselines on are shown in the left part of Table 2. It is shown that the data-mix model outperforms the ensemble approach by a large margin. Therefore, we base our pruning experiments on the data-mix model.

Following insights from the last section, we then prune out data from the large-version language models (i.e., GPT4, GPT-Neo2.7B, LLaMA13B and GPT-2xl) and train a detector by mixing up training data from the remaining generators. [5] The degree of drop on the worst-case accuracy reflects the zero-shot generalization ability of the proposed detector.

Also shown in Table 2, the accuracy of the proposed detectors (both average and worst-case) remains similar to or only slightly decreases compared to the data-mix baseline. Figure 2 provides detailed information on the changes in accuracy after pruning out four large-version models. The accuracy of the proposed detector only experiences a slight decrease (<3%) for GPT4 and LLaMA13B. **Our results show that in the case of limited budget or computing, data from the medium-version LM can decently approximate the large-version in an ensembled data collection.**

On the right part of Table 2, we conduct com-

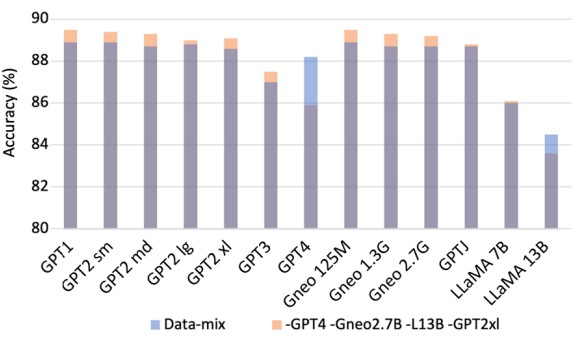

Figure 2: Accuracy comparison on each generator before and after pruning out four large-version LLMs on the RealNews dataset.

parison experiments where both medium and large versions are pruned out. As expected, this results in a worse performance on detection of the pruned generators, reflected by the worst-case accuracy. Especially, the comparison experiment of pruning out both GPT3 and GPT4 is quite alarming: The detector trained from combined data of all other generators only has accuracy around 42% (RealNews) or 62% (IMDBReview). This implies that if OpenAI did not give public access to generations of the two models, existing detectors would fail.

## 5 Related Work

We now discuss the literature most related to our work, and defer a more complete review to Appendix A. Pagnoni et al. (2022a) demonstrate the degraded performance of trained detectors under different threat scenarios, while the range of generator models is not as wide or up-to-date as our work. Liang et al. (2023) study the bias of detectors for LLMs in the case of non-native English writers. In a very recent and concurrent work, Mireshghallah et al. (2023) study the generalization of detectors under the DetectGPT (Mitchell et al., 2023) algorithm, which is also shown to be far from perfect. Comparing to a trained detector, DetectGPT relies on access to the generator LLM, which might be expensive.

## 6 Conclusion and Discussion

In this work, we observe a generalization relationship among detectors trained on different generators in three domains, where detectors for medium-version models demonstrate the ability to effectively generalize to the larger-version. Building upon this finding, we prune out data from large-version generators in an ensembled training dataset and demonstrate that the performance loss is min-

---

[4]We have also tried another baseline where we average parameters from all detectors. However, the performance is worse than the majority-voting baseline, and we omit this result.

[5]Our data collection for GPT4 costs around $450.

imal. Our results indicate that practitioners with limited budget or computing resources can use data from medium-size LLMs as a good approximation for the large version.

With the rapid release of various LLMs and generation APIs, a detector needs to cover a wide range of generators. While our work makes some initial progress, our experiments show that the detection of an unseen (or non-public) generator is still a difficult and open question. We hope our work could motivate more research devoted to this important direction.

## Limitations

Our work focuses on supervised detector models and there are other approaches for machine-generated text detection (Appendix A). In a very recent and concurrent work, Mireshghallah et al. (2023) studies the generalization of detectors under the DetectGPT (Mitchell et al., 2023) algorithm, which is also shown to be far from perfect. Comparing to a trained detector, DetectGPT relies on access to the generator LLM, which might be expensive. It is also interesting to base the detector on a larger LM than ELECTRA-large, but we surmise the observations should be similar.

The zero-shot generalization ability of detectors shown in this work implies that different generators are making similar artifacts based on which the detectors make decisions. As future work, it would be interesting to examine the salient features (Zeiler and Fergus, 2014) and compare between machine/human-generated text.

Finally, our experiments show that the detection of an unseen or non-public generator is still a difficult and open question. For example, the combination of data from all other generators can not generalize to GPT3 and GPT4. This important research direction deserves more research efforts.

## Ethics Statement

The detection of machine-generated text has important applications such as detecting fake news and fake reviews on the internet. However, it could also introduce new risks: Malicious parties can use released detectors to develop text generation systems that evade existing detectors in an adversarial manner. Our experiments show that the detection of an unseen (or non-public) generator is still a difficult and open question, and we hope our work could motivate more research devoted to this important

direction.

## Acknowledgements

We thank the reviewers, the area chair, Prof. Xiaojun Wan, Hao Wang, Shangbin Feng, and Yichen Wang. We gratefully acknowledge support from NSF CAREER Grant No. IIS2142739. This material is funded in part by the DARPA Grant under Contract No. HR001120C0124. This research is supported in part by the Office of the Director of National Intelligence (ODNI), Intelligence Advanced Research Projects Activity (IARPA), via the HIATUS Program contract #2022-22072200004. The views and conclusions contained herein are those of the authors and should not be interpreted as necessarily representing the official policies, either expressed or implied, of ODNI, IARPA, or the U.S. Government. The U.S. Government is authorized to reproduce and distribute reprints for governmental purposes notwithstanding any copyright annotation therein.

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

## Supplemental Materials

## A   Related Work

Research on detecting machine-generated text can be roughly divided into two categories: supervised training and zero-shot detection (To clarify, in the literature "zero-shot" usually means that the approach does not require training data, while our work focus on zero-shot generalization to the detection of a held-out generator).

In the case of supervised methods, Bakhtin et al. (2019) train an energy-based model to identify machine-generated text. Zellers et al. (2020) trainn a GROVER detector and finds that models exhibiting superior performance in generating neural disinformation are also highly effective in detecting their own generated content. Both Solaiman et al. (2019) and Ippolito et al. (2020) propose zero-shot approaches to detect machine-generated text and evaluate the capability of pretrained models. Liu et al. (2022) present a coherence-based contrastive learning model to detect the machine-generated text under low-resource scenario. Kirchenbauer et al. (2023) propose a watermarking method (Abdelnabi and Fritz, 2021) which introduces designed noise which is imperceptible to human readers. Mitchell et al. (2023) propose DetectGPT, a zero-shot method that utilizes a novel curvature-based criterion to determine whether a text is generated by a specific model. This approach has demonstrated superior detection capabilities compared to other existing zero-shot methods. While DetectGPT does not require training a separate detector, it relies on access to the generator LLM, which can be costly. Recently, Su et al. (2023) follow up the work of DetectGPT and introduce two new zero-shot methods: DetectLLM-LRR and DetectLLM-NPR.

## B   Auxiliary Results

In Figure 3, we plot detector-generator pairs with large (>20%) Acc-Gap on the three datasets. It shows that none of detectors is able to generalize to all generators. For example, all detectors except $D_{\text{GPT4}}$ has large accuracy gap for GPT3.

In Figure 4, 5 and 6, we give detailed heatmaps of Acc-Gap for every detector/generator pair on the three datasets. The reported numbers are calculated as the averages of Acc-Gap obtained by bootstrapping 100 times.

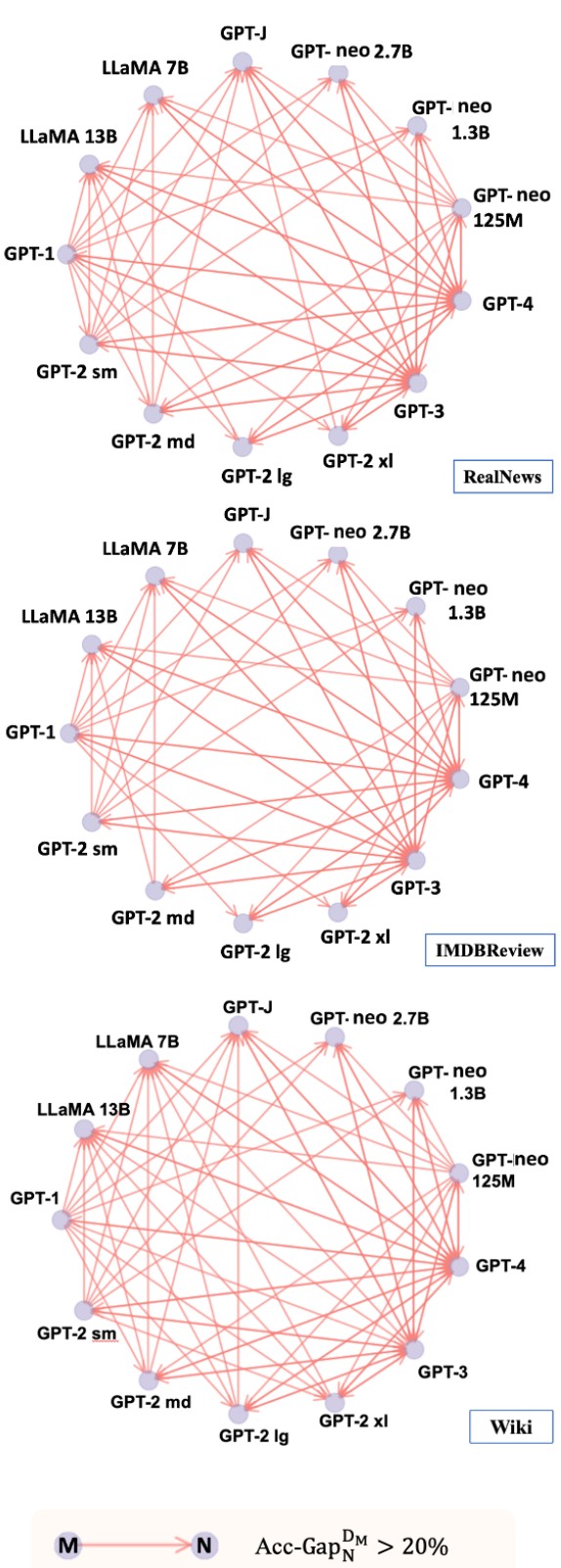

Figure 3: Detector-generator pairs with large (>20%) accuracy gap.

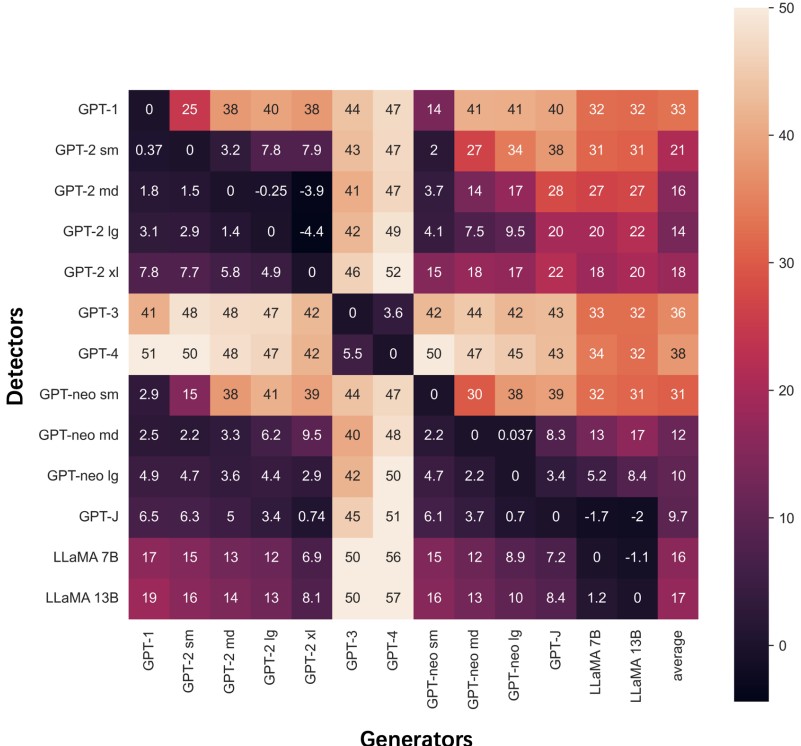

Figure 4: Heapmap of Acc-Gap for detector/generator pairs on the RealNews dataset.

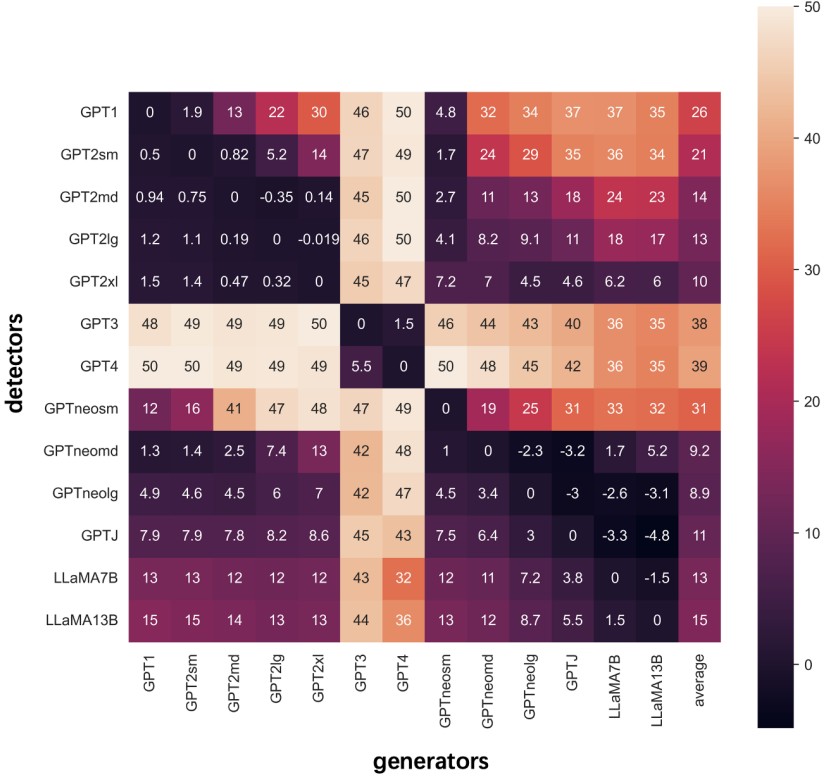

Figure 5: Heapmap of Acc-Gap for detector/generator pairs on the IMDBReview dataset.

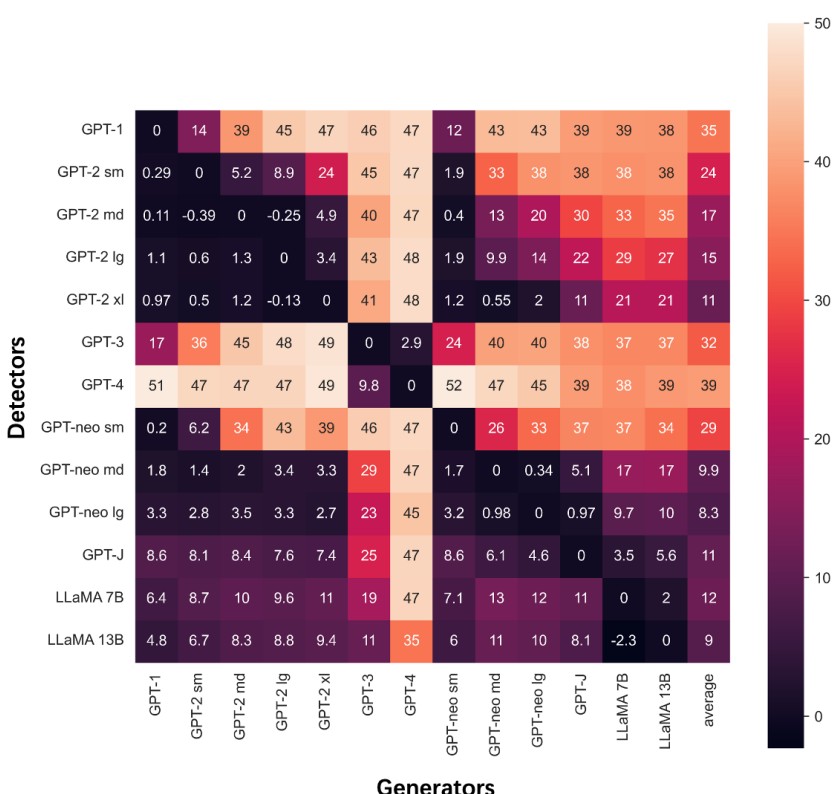

Figure 6: Heapmap of Acc-Gap for detector/generator pairs on the Wikipedia dataset.