# OpenReview forum: "On the Zero-Shot Generalization of Machine-Generated Text Detectors"
_EMNLP/2023/Conference — EMNLP 2023 Findings_

### Official Review · Reviewer_GhLs · 2023-07-28

**Soundness:** 2

**Excitement:**

2: Mediocre: This paper makes marginal contributions (vs non-contemporaneous work), so I would rather not see it in the conference.

**Paper Topic And Main Contributions:**

This paper discusses a task related to distinguishing text generated by a language model and text written by humans. Specifically, it examines whether a detector can effectively generalize when using text generated by different decoder-based language models during training and testing. The study discovers that a medium-sized language model can also detect text produced by a large language model effectively. Based on this finding, the paper proposes a method to train the detector by excluding text generated by the large language models (e.g., GPT4) from the set of texts generated by different language models. The pruning method (i.e., data exclusion) detects AI generated text better than other baseline ensemble techniques. However, personally, I find the experimental setup to be insufficiently concrete and practical to draw the same conclusions as the authors (see the weakness).

**Questions For The Authors:**

NA

**Reasons To Accept:**

- Conducted diverse and rigorous experiments using diverse language models with varying size.
- Contrary to the intuitive belief that texts generated by larger language models would be more difficult to discriminate, this paper discovered a counter-intuitive observation.
- Based on this finding, the paper exhibited simply removing text generated from large language model can help detecting AI generated text from human writing.

**Reasons To Reject:**

Personally, the most concerning aspect of this paper, in my view, is the lack of concrete experimentation settings. I have the following negative opinions about this setup:

- Human writing data & its distribution: The distribution of  human writing is highly diverse. While the authors categorized the dataset into domains such as news, knowledge, and reviews, even within each domain, there can be a wide variety of writing styles. Approximating this complex and challenging distribution with 5000 samples from a single dataset can be highly biased and is likely to represent only a small portion of the entire human writing distribution. In such cases, it is unclear whether the performance in the paper can be seen as the actual performance of AI-generated detection.
- Language model text generation: When using Language Models (LLM) to generate text, it has become a common paradigm to provide various prompts. Specifically, the capability of large models like GPT-4 or ChatGPT comes from following such complex instructions. However, in this paper, the authors adopt a naïve method of purposelessly auto-generating from the initial 20 truncated tokens. So, I believe drawing conclusions that small models can sufficiently detect text generated by large models is quite misleading.
- Binary detector's ability: In this paper, the two aforementioned datasets are used to train a binary classifier, which is then used as a AI generated text detector. However, I believe that such a trained detector does not possess the ability to reliably distinguish between human language and machine-generated language. Specifically, when distinguishing between the two types of text collected earlier, there could be various cues or biases that could assist in making the distinction. For instance, the human news data may be limited to news-related semantics, while the text generated by the machine could cover various semantics, making it easier to differentiate between the two. Neural models are known to utilize various unintended cues (i.e., shortcut cues) for learning in such scenarios.

**Reproducibility:**

4: Could mostly reproduce the results, but there may be some variation because of sample variance or minor variations in their interpretation of the protocol or method.

**Reviewer Confidence:**

4: Quite sure. I tried to check the important points carefully. It's unlikely, though conceivable, that I missed something that should affect my ratings.

---

> ### Author Rebuttal · Authors · 2023-08-29
>
> Thank you for the comments and we would like to provide explanation and counterarguments regarding the points the reviewer has raised.
>
> The reviewer questions the way we prompt LLMs to collect samples from the instruction tuned models, i.e. by truncating the initial 20 tokens from the human samples as input prompts for LLMs. For our experiments involving the instruction-tuned GPT-4 and GPT-3, however, **we do employ a specific prompt**, "Please continue this text in about 90 words:", preceding the initial 20 tokens to guide the models in generating coherent text. This minor implementation detail is omitted in the current version because it is not critically relevant to the main research questions and their corresponding experiments in the paper, and due to the space limits. We will incorporate this detail into the main text or as an appendix.
>
> Furthermore, our pipeline to use prefix from human-written samples as input prompts for text completion in the data preparation process simply follows relevant prior work, e.g. "Threat Scenarios and Best Practices to Detect Neural Fake News" [1], "Can AI-Generated Text be Reliably Detected?" [2] and "Red Teaming Language Model Detectors with Language Models" [3].
>
> Importantly, we would like to clarify that **our intent is not to showcase the usage of prompts**; Rather than how to do prompting, our focus is on the detection of the generation of different models.
>
>
> To the comment on **"Human writing data \& its distribution"** and **"Binary detector's ability"**:
>
> The reviewer argues that selecting only one or a few datasets may not adequately represent all human writing styles. While this point seems valid, please note that our data collection pipeline follows common practices as the related works (also discussed above). The distribution of human samples is not the focus of our work, instead, the focus and findings of our paper center around **the generalization patterns observed among detectors trained on different LLMs' generations**. We experiment on datasets from different domains and the consistency in generalization patterns across all three domains lead us to believe that these findings are consistent and not domain-specific. **While the concern raised by the reviewer regarding potential bias in human samples is valid, it does not impact our conclusions and is not the focus of this work.**
>
>
> [1] Pagnoni A, Graciarena M, Tsvetkov Y. Threat scenarios and best practices to detect neural fake news[C]//Proceedings of the 29th International Conference on Computational Linguistics. 2022: 1233-1249.
>
> [2] Sadasivan V S, Kumar A, Balasubramanian S, et al. Can ai-generated text be reliably detected?[J]. arXiv preprint arXiv:2303.11156, 2023.
>
> [3] Shi Z, Wang Y, Yin F, et al. Red Teaming Language Model Detectors with Language Models[J]. arXiv preprint arXiv:2305.19713, 2023.

---

### Official Review · Reviewer_wjLU · 2023-08-04

**Soundness:** 4

**Excitement:**

4: Strong: This paper deepens the understanding of some phenomenon or lowers the barriers to an existing research direction.

**Paper Topic And Main Contributions:**

This study aims to solve the problem of identifying differences between machine-generated texts and genuine human texts. The main idea is to generate data from different models and then train neural detectors on data from each generator and test its performance on other generators. Two baselines are compared: model ensemble, where we aggregate predictions from all detectors by majority voting or conﬁdence (probability) average and data mixing, where we train a new ELECTRA-large detector by mixing up the training data from all generators. There are also experiments with prune out data from large version generators in an ensembed training dataset.

**Reasons To Accept:**

This study examines a very relevant topic at the moment related to the development of large language models (i.e., GPT 4, GPT-Neo2.7B, LLaMA13B and GPT-2xl) and using them for base NLP tasks. A positive result, namely, this study shows that in the case of limited budget or computing, data from the medium-version LM can decently approximate the large-version in an ensembled data collection, show the usefulness of the study in the development of approaches for identifying machine-generated texts. Report on the work of the two considered baselines (model ensemble, data mixing) can further help in the development and improvement of models for solving the current problem.

**Reasons To Reject:**

The authors of the article did not explain why the choice of the detector model fell on ELECTRA-large, I would like to see a comparison with other models for detector in this task. It is necessary to provide for comparison any other approaches to solve this problem, the paper presents only the baselines implemented by the authors. As an important result, the researchers cite the possibility for the medium-version LLMs generalize to the large-version models. In this regard, it is interesting to look at the result using distilled model for both LLM and detector. It makes sense to conduct an experiment where the ensemble of detectors consists of distilled models corresponding to each of the LLMs selected for generating training data.

**Reproducibility:**

3: Could reproduce the results with some difficulty. The settings of parameters are underspecified or subjectively determined; the training/evaluation data are not widely available.

**Reviewer Confidence:**

3: Pretty sure, but there's a chance I missed something. Although I have a good feel for this area in general, I did not carefully check the paper's details, e.g., the math, experimental design, or novelty.

---

> ### Author Rebuttal · Authors · 2023-08-29
>
> We appreciate that the reviewer took the time to carefully review our paper and shared interesting suggestions.
>
>
> >  The authors of the article did not explain why the choice of the detector model fell on ELECTRA-large,
>
> Because prior related work [1] shows ELECTRA has strong performance as a detector.
>
>
>
> > I would like to see a comparison with other models for detector in this task.
>
> Per the Reviewer's request, we experiment with an additional base detector, namely Albert [2]. While the absolute accuracies are slightly different, we find that the key observation---that the detectors trained for the medium-size models can generalize to larger-size models---still holds, as shown in Table 1. This experiment shows that the results are not Electra-specific, but are more general. We will add these experiments and discussion in the final revision of the paper.
>
>
> | M       | N       | **$\text{Gap}^{D_M}_N$** | **$\text{Gap}^{D_N}_M$** |
> | ------- | ------- | ------------------------ | ------------------------ |
> | GPT3    | GPT4    | 1.22\%                   | 1.60\%                   |
> | LLa7B   | LLa13B  | -0.22\%                  | 0.17\%                   |
> | Neo1.3B | Neo2.7B | -0.13\%                  | 0.09\%                   |
> | GPT2lg  | GPT2xl  | -1.44\%                  | 1.70\%                   |
> Table 1. We train detectors of generated text based on the Albert LM in the news domain, and evaluate their capacity to detect machine-generated text from PLMs (listed in col. M and N). ${Gap}^{D_M}_N$ denotes the drop of accuracy rate when the detector is trained on generator M (medium-version model) instead of N(large-version model) itself, and ${Gap}^{D_N}_M$ denotes the accuracy drop in the opposite direction. When detector is based on ALBERT Large v2, we have an observation similar to when the detector is based on Electra, i.e., the generalization performance is slightly worse than in the other direction, reflected by a larger Acc-Gap.
>
>
> The Reviewer also suggests trying the distilled versions for both the LLM and the detector, which we agree is a good and interesting idea. Although this falls outside the scope of our current study which focuses on the range of existing LLMs, we will add this into our discussion of future work and we will acknowledge the Reviewer, thanks!
>
>
> [1] Pagnoni A, Graciarena M, Tsvetkov Y. Threat scenarios and best practices to detect neural fake news[C]//Proceedings of the 29th International Conference on Computational Linguistics. 2022: 1233-1249.
>
> [2] Lan, Zhenzhong, et al. "ALBERT: A Lite BERT for Self-supervised Learning of Language Representations." (2020). In Proc. ICLR

---

### Official Review · Reviewer_FNgs · 2023-08-05

**Soundness:** 3

**Excitement:**

4: Strong: This paper deepens the understanding of some phenomenon or lowers the barriers to an existing research direction.

**Paper Topic And Main Contributions:**

This paper investigates whether the detectors for model/machine-generated text work across different text generators in a zero-shot manner (i.e. the different text generators' generations are not in the training set). They find that generalization is hard, but detectors trained on outputs of medium-sized models generalize reasonably well to larger ones. They use this insight to build detectors that are fairly robust by using an ensemble of these medium-sized models.

**Questions For The Authors:**

- l114; What wikipedia dataset?

Thank you authors for the rebuttal and answering the questions I had. I will raise my excitement score to a 4 based on some of the empirical results and additional experiments that shed light on the consistency of the method.

**Reasons To Accept:**

1) This problem is well-motivated and the research question is an important one

2) The study is nicely scoped and doesn't try and do too much.

3) The addition of bootstrapping here is good to establish variance and be able to conduct a t-test.

**Reasons To Reject:**

1) I think the experiments could be expanded. The choice of classifier, the amount of training data (4000 machine-generated samples in train per generator), and the data types could be varied.

- I imagine the choice of classifier has an impact here in that a different architecture could bias towards generations of a certain type or models or a certain type. You discuss scaling up the detector in the limitations section, but it'd be nice to see a larger array of things.
- I imagine the data limitation is a big one. A classifier with just a few examples is unlikely to be robust at all, whereas for such a large model like you're using a larger classifier could utilize much more data to get a better fit on the distribution for the generator at hand.
- The data types are a mystery to me, but its conceivable that datasets that have been studied and designed to be prompts could service you well here (e.g. RealToxicityPrompts)

2) The Datasets section could be better described. I have no idea what aspect of wikipedia you took data from. I don't really know whether you sample text at a document level and then truncate to create your data for the discriminator etc.

I understand that this is a short paper, so the scope of experiments and contribution is less than a long paper, but some additional clarity on these facets would go a long way to improving the paper.

**Reproducibility:**

4: Could mostly reproduce the results, but there may be some variation because of sample variance or minor variations in their interpretation of the protocol or method.

**Reviewer Confidence:**

3: Pretty sure, but there's a chance I missed something. Although I have a good feel for this area in general, I did not carefully check the paper's details, e.g., the math, experimental design, or novelty.

**Typos Grammar Style And Presentation Improvements:**

- The results section is a little confusing to read through and I had to read through it a few times to understand what's going on, so a rewrite to be a little crisper and clearer here would be wonderful.

---

> ### Author Rebuttal · Authors · 2023-08-29
>
> We appreciate the insightful comments from the Reviewer. We would like to provide responses for the "reasons to reject" and "Questions For The Authors" sections:
>
> > - I imagine the choice of classifier has an impact here in that a different architecture could bias towards generations of a certain type or models or a certain type. You discuss scaling up the detector in the limitations section, but it'd be nice to see a larger array of things.
>
> Per the Reviewer's request, we experiment with an additional base detector, namely Albert [1]. While the absolute accuracies are slightly different, we find that the key observation---that the detectors trained for the medium-size models can generalize to larger-size models---still holds, as shown in Table 1. This experiment shows that the results are not Electra-specific, but are more general. We will add these experiments and discussion in the final revision of the paper.
>
>
> | M   | N   | **$\text{Gap}^{D_M}_N$** | **$\text{Gap}^{D_N}_M$** |
> | --- | --- | -------------------- | ---------------------- |
> |GPT3    | GPT4    | 1.22\%  | 1.60\% |
> |LLa7B   | LLa13B  | -0.22\% | 0.17\% |
> |Neo1.3B | Neo2.7B | -0.13\%  | 0.09\% |
> |GPT2lg  | GPT2xl  | -1.44\% | 1.70\% |
> Table 1. We train detectors of generated text based on the Albert LM in the news domain, and evaluate their capacity to detect machine-generated text from PLMs (listed in col. M and N). ${Gap}^{D_M}_N$ denotes the drop of accuracy rate when the detector is trained on generator M (medium-version model) instead of N(large-version model) itself, and ${Gap}^{D_N}_M$ denotes the accuracy drop in the opposite direction. When detector is based on ALBERT Large v2, we have an observation similar to when the detector is based on Electra, i.e., the generalization performance is slightly worse than in the other direction, reflected by a larger Acc-Gap.
>
>
> >  - I imagine the data limitation is a big one. A classifier with just a few examples is unlikely to be robust at all, whereas for such a large model like you're using a larger classifier could utilize much more data to get a better fit on the distribution for the generator at hand.
>
> Per the reviewer's request, we expand the training samples (from 4000 machine-generated samples per generator in the training set to 9000). While the average accuracy of all detectors (when tested on the same generator on which they were trained) has increased from 94.1\% to 96.9\%, we got a consistent observation as the main findings in our paper: the detector trained on a medium-size model demonstrates a noteworthy ability to generalize to the larger-size model. We present the additional experimental results in Table 2, and we will add this discussion in the next revision.
>
>
> | M   | N   | $\text{Gap}^{D_M}_N$ | $\text{Gap}^{D_N}_M$ |
> | --- | --- | -------------------- | -------------------- |
> |GPT3    | GPT4    | 3.09\%  | 6.23\% |
> |LLa7B   | LLa13B  | -2.66\% | 3.62\% |
> |Neo1.3B | Neo2.7B | 0.47\%  | 1.07\% |
> |GPT2lg  | GPT2xl  | -2.48\% | 3.49\% |
> Table 2. We train detectors of generated text on the expanded dataset in the news domain, and evaluate their capacity to detect machine-generated text from PLMs (listed in col. M and N).${Gap}^{D_M}_N$ denotes the drop of accuracy rate when the detector is trained on generator M (medium-version model) instead of N(large-version model) itself, and ${Gap}^{D_N}_M$ denotes the accuracy drop in the opposite direction. We find that when attempting to generalize from the large-version models to medium ones, the generalization performance is slightly worse than in the other direction, reflected by a larger Acc-Gap.
>
>
> > - The data types are a mystery to me, but its conceivable that datasets that have been studied and designed to be prompts could service you well here (e.g. RealToxicityPrompts)
>
> Our data collection pipeline follows common practices as the related work, including "Threat Scenarios and Best Practices for Neural Fake News Detection" [2], "Can AI-Generated Text be Reliably Detected?" [3] and "Red Teaming Language Model Detectors with Language Models" [4], using partial human samples as input prompts to the language model for text completion, thereby constructing the dataset.
>
>
> >  2. The Datasets section could be better described. I have no idea what aspect of wikipedia you took data from. I don't really know whether you sample text at a document level and then truncate to create your data for the discriminator etc.
>
> and
>
> >  l114; What wikipedia dataset?
>
> The Wikipedia dataset we used was directly obtained from Hugging Face, data subset "20220301.en". This particular dataset is a popular resource available on the Hugging Face platform (page link: https://huggingface.co/datasets/wikipedia). We add the citation for this dataset in our paper (line 350) according to the "Citation Information". We will incorporate the Hugging Face webpage link into the citation for this dataset in the next revision.
>
>
> [1] Lan, Zhenzhong, et al. "ALBERT: A Lite BERT for Self-supervised Learning of Language Representations." (2020). In Proc. ICLR
>
> [2] Pagnoni A, Graciarena M, Tsvetkov Y. Threat scenarios and best practices to detect neural fake news[C]//Proceedings of the 29th International Conference on Computational Linguistics. 2022: 1233-1249.
>
> [3] Sadasivan V S, Kumar A, Balasubramanian S, et al. Can ai-generated text be reliably detected?[J]. arXiv preprint arXiv:2303.11156, 2023.
>
> [4] Shi Z, Wang Y, Yin F, et al. Red Teaming Language Model Detectors with Language Models[J]. arXiv preprint arXiv:2305.19713, 2023.

---

### Meta-Review · Area_Chair_bSwe · 2023-09-23

**Recommendation:** 3

**Metareview:**

Reviewers on the whole agree on the soundess of the work, and it's excitement. While the points raised by reviewer GhLs on `Human writing data & its distribution` are correct, the paper's focus is on whether or not detectors can identify the data from generators they have not seen data for previously.  Admitedly while this is a closely related question - a person with a new wrting style is akin to a new generator, the authors have followed the methodology of recent work in the area.  It would of been good the challenge some of these practices as the authors know their limitations.

---

### Decision · Program_Chairs · 2023-10-07

**Decision:**

Accept-Findings

**Comment:**

Reviewers on the whole agree on the soundess of the work, and it's excitement. While the points raised by reviewer GhLs on `Human writing data & its distribution` are correct, the paper's focus is on whether or not detectors can identify the data from generators they have not seen data for previously.  Admitedly while this is a closely related question - a person with a new wrting style is akin to a new generator, the authors have followed the methodology of recent work in the area.  It would of been good the challenge some of these practices as the authors know their limitations.